# Comparison between Laparoscopic and Robotic Approach for Sentinel Lymph Node Biopsy in Endometrial Carcinoma Women

**DOI:** 10.3390/jpm13010029

**Published:** 2022-12-23

**Authors:** Antonio Raffone, Diego Raimondo, Arianna Raspollini, Alessia Oliviero, Antonio Travaglino, Federica Renzulli, Giulia Rovero, Simona Del Forno, Gabriella Vullo, Antonio Simone Laganà, Vito Chiantera, Renato Seracchioli, Paolo Casadio, Antonio Mollo

**Affiliations:** 1Department of Medical and Surgical Sciences (DIMEC), University of Bologna, 40138 Bologna, Italy; 2Division of Gynaecology and Human Reproduction Physiopathology, IRCCS Azienda Ospedaliero-Universitaria di Bologna, 40138 Bologna, Italy; 3Gynecology and Obstetrics Unit, Department of Medicine, Surgery and Dentistry Schola Medica Salernitana, University of Salerno, 84081 Baronissi, Italy; 4Unità di Ginecopatologia e Patologia Mammaria, Dipartimento Scienze della Salute della Donna, del Bambino e di Sanità Pubblica, Fondazione Policlinico Universitario A. Gemelli IRCCS, 00168 Rome, Italy; 5Clinica Mediterranea Hospital, 80122 Napoli, Italy; 6Unit of Gynecologic Oncology, ARNAS “Civico-Di Cristina-Benfratelli”, Department of Health Promotion, Mother and Child Care, Internal Medicine and Medical Specialties (PROMISE), University of Palermo, 90127 Palermo, Italy

**Keywords:** ICG, indocyanine green, lymphadenectomy, mapping, treatment, lymph node dissection, detection, staging, minimally invasive surgery, endoscopy

## Abstract

Robotic surgery has been approved as an alternative to laparoscopy to improve surgical outcomes. There is neither a consensus nor a systematic assessment of the literature about the superiority of the robotic approach over the laparoscopic one for sentinel lymph node (SLN) biopsy in endometrial carcinoma (EC) women. Therefore, a systematic review and meta-analysis was performed to compare the laparoscopic and robotic approaches for SLN biopsy in EC patients. Five electronic databases were queried from their inception to May 2022 for peer-reviewed studies, comparing such approaches in SLN biopsy in EC patients. The rate of detected SLN, dissected SLN, intraoperative and postoperative complications, conversion to laparotomy, number of dissected SLN, and SLN identification and dissection time were compared between the laparoscopic and robotic approaches for SLN biopsy in EC patients. Odds ratios with 95% confidence intervals were calculated when possible. Two studies with 660 EC women (364 who had undergone laparoscopy, and 296 who had robotic surgery) were included. No assessed outcome showed significant differences between the two approaches. In conclusion, the laparoscopic and robotic approaches for SLN biopsy in EC patients appeared to not differ, in terms of SLN detection, intraoperative and postoperative complications, conversion to laparotomy, number of dissected SLN, and SLN identification and dissection time.

## 1. Introduction

Endometrial carcinoma (EC) is the most common gynecological cancer in the western world, with an even higher increase in number of deaths than in incidence in the last 20 years [1,2,3,4,5]. For early-stage ECs, total hysterectomy with bilateral salpingo-oophorectomy and nodal staging, including pelvic lymphadenectomy, with or without para-aortic lymphadenectomy, is considered the standard of treatment [6,7]. For the staging of early-stage ECs, the National Comprehensive Cancer Network (NCCN) guidelines agreed on sentinel lymph node (SLN) biopsy as an alternative to systematic lymphadenectomy, in order to reduce the risk of long-term complications and morbidity [6,8]. Moreover, a recent systematic review demonstrated that, even in high-risk groups of early-stage EC patients, SLN biopsy can be adopted [9]. In order to detect SLN, near infrared radiation imaging after injection of indocyanine green (NIR-ICG) is required with laparoscopic or robotic-assisted surgical approaches.

In fact, due to their minimal invasiveness, the role of these approaches has largely increased over the last few decades [10,11]. Robotic surgery was approved for gynecological surgery by the Food and Drug Administration in 2005, with the aim of improving several surgical interventions, due to a better visualization through 3D imaging, a more precise control of the instrumentation, and a better ergonomics for the surgeons [12,13]. In fact, several studies compared robotic and laparoscopic approaches in different surgical interventions, reporting conflicting findings [14,15,16,17,18]. On the other hand, in some categories of patients, such as obese women, robotic surgery has largely shown perioperative advantages and a lower conversion rate, compared to laparoscopy [19,20,21,22,23].

On these bases, although we might hypothesize that robotic surgery might have better surgical outcomes than laparoscopy, even in SLN biopsy, there is no consensus and a systematic assessment of available literature is lacking to date [24,25].

The aim of this systematic review and meta-analysis was to compare laparoscopic and robotic approaches for SLN biopsy in EC patients.

## 2. Materials and Methods

### 2.1. Study Protocol and Reporting

The study followed an a priori defined study protocol and was reported, following the preferred reporting items for systematic reviews and meta-analyses (PRISMA) statement [26]. Three authors independently concluded each review stage, discussing disagreements with senior authors.

### 2.2. Search Strategy and Study Selection

Five electronic databases (i.e., Google Scholar, Web of Sciences, ClinicalTrial.gov, Scopus, and MED-LINE) were queried, from their inception to May 2022, for peer-reviewed studies comparing laparoscopic and robotic approach in SLN biopsy in EC patients. Studies in languages other than English, video articles, literature reviews, and case reports were a priori considered as exclusion criteria.

In particular, the following text words were searched in different combinations: ‘’indocyanine green’’; ‘’ICG’’; ‘’fluorescence’’; ‘’firefly’’; “minimally invasive”; ‘’laparoscop*’’; “robotic*”; “route”; “approach”; ‘’gynecol*’’; “gynaecol’’; ‘’uter’’; ‘’endometr’’; ‘’cancer’’; “tumour”; ‘’tumor’’; ‘’carcinoma’’; ‘’neoplasia’’; ‘’malignanc*’’; “SLN”; “sentinel”; “lymph node”; “lymphadenectomy”; “dissection”; “biopsy”. In addition, references from each eligible article were also screened for relevant studies.

### 2.3. Risk of Bias within Studies Evaluation

The risk of bias within the included studies was independently evaluated by three authors via the methodological index for non-randomized studies (MINORS) [27].

In particular, the following seven applicable domains were examined for each study: (1) Aim (i.e., was the question addressed in a precise and relevant way?); (2) Patient selection (i.e., were all the eligible patients included during the study period?); (3) Data collection (i.e., were all data collected according to the protocol established before the beginning of the study?); (4) Endpoints (i.e., were all endpoints appropriate for the aim of the study?) (5) Unbiased endpoints evaluation (i.e., was the evaluation of the study endpoints unbiased?); (6) Follow-up (i.e., was the follow-up enough to assess the endpoints?); (7) Loss to follow-up (i.e., was the patient loss to follow-up less than 5%?).

The included studies were judged by the authors at “low risk”, “high risk”, or “unclear risk” of bias based on data were “reported and adequate”, “reported but inadequate”, or “not reported”, respectively.

### 2.4. Data Extraction

The PICO (population, intervention or risk factor, comparator, outcomes) items [26] were adopted for data extraction. In details, “Population” was women with EC; “Intervention” was the laparoscopic approach for SLN biopsy; “Comparator” was robotic approach for SLN biopsy; “Outcome” was: SLN detection rate, number of dissected SLN, SLN identification time, SLN dissection time, intraoperative complications rate, postoperative complications rate, and conversion to laparotomy rate. In particular, detection rate was separately assessed for bilateral, unilateral, and overall detection.

### 2.5. Data Synthesis

The rate of detected SLN, dissected SLN, intraoperative and postoperative complications, and conversion to laparotomy were compared between laparoscopic and robotic approach for SLN biopsy in EC patients calculating odds ratio (OR). In particular, OR was assessed for as individual and pooled estimates, and it was reported on forest plots, with 95% confidence intervals (CI). Statistical significance of OR values was evaluated using the Z-test with a significant *p*-value < 0.05.

The inconsistency index I^2^ was adopted for estimating statistical heterogeneity among included studies, as follows: heterogeneity null when I^2^ = 0%, insignificant when 0% < I^2^ ≤ 25%, low when 25% < I^2^ ≤ 50%, moderate when 50% < I^2^ ≤ 75%, and high when I^2^ > 75%, as previously described [28,29,30].

The random effect model of DerSimonian-Laird was adopted for data synthesis.

Data synthesis was performed through Review Manager 5.4 (Copenhagen: The Nordic Cochrane Centre, Cochrane Collaboration, 2014).

## 3. Results

### 3.1. Study Selection and Characteristics

Through database searches, 1178 articles were identified. Of them, 192 articles remained after duplicate removal, and 51 remained after title screening, while 8 articles were evaluated for eligibility after abstract screening. Lastly, two studies were included in the systematic review and meta-analysis [24,25] (Figure 1).

Both included studies were designed as observational, retrospective, and cohort studies, and we assessed a total of 660 EC women, 364 who had undergone laparoscopy, and 296 who had undergone robotic surgery (Table 1).

Patient age ranged from 28 to 88 years in the laparoscopic group and from 25 to 84 in the robotic group, while body mass index ranged from 16.7 to 50 kg/m^2^ in the laparoscopic group and from 18.7 to 64.1 kg/m^2^ in the robotic surgery group (Table 2).

Histologic features of ECs are reported in Table 2, while details about indocyanine green injection and lymph node detection and mapping are reported in Table 3.

### 3.2. Risk of Bias within Studies Assessment

All the included studies were judged by the low risk of bias for the following domains: “Aim”, “Patient selection”, “Data collection”, and “Unbiased study endpoints evaluation”. The study by Bizzarri et al. [25] was also judged by the low risk of bias for the remaining domains.

On the other hand, the study by Chaowawanit et al. [24] was judged by the risk of bias for the following domains:-“Endpoints” (unclear risk) and “Follow-up” (high risk) because they did not assess intraoperative and postoperative complications with related necessary follow-up.-“Loss to follow-up” (unclear risk) because patients lost to follow-up were more than 5% of the whole study population.

Risk of bias within studies evaluation is graphically shown in Figure 2.

### 3.3. Data Synthesis

Compared to laparoscopy, robotic surgery for SLN biopsy in EC women showed ORs of:1.80 [95%CI: 0.35, 9.17] for overall detection (Figure 3);1.12 [95%CI: 0.56, 2.23] for bilateral detection (Figure 4);1.12 [95%CI: 0.45, 1.67] for unilateral detection (Figure 5);1.22 [95%CI: 0.75, 1.96] for number of dissected SLN: 1 (Figure 6);1.06 [95%CI: 0.76, 1.48] for number of dissected SLN: 2 (Figure 7);0.99 [95%CI: 0.65, 1.51] for number of dissected SLN: 4 (Figure 8);1.85 [95%CI: 0.17, 20.47] for number of dissected SLN: 6 (Figure 9);0.92 [95%CI: 0.18, 4.59] for intraoperative complications (Figure 10);0.37 [95%CI: 0.13, 1.07] for postoperative complications (Figure 11);3.76 [95%CI: 0.79, 17.85] for conversion to laparotomy (Figure 12).

The number of dissected SLN, SLN identification time, and SLN dissection time were not pooled because of the non-extractable data from the included studies. In particular, the number of dissected SLN was reported as median and range, while the SLN identification was reported as mean and range (Table 2). However, not significant differences in such outcomes were found in the primary analyses of the included studies.

## 4. Discussion

This study shows that there are no differences between laparoscopic and robotic approach for SLN biopsy in EC patients. In particular, SLN detection, intraoperative and postoperative complications, conversion to laparotomy, number of dissected SLN, and SLN identification, and dissection time do not differ among the surgical approaches.

In 2005, robotic surgery has been approved by the Food and Drug Administration as a minimally invasive approach, alternative to laparoscopy for gynecological surgical purposes [31]. Such a novel approach aims to bring surgical advantages because of better visualization, ergonomics, and instrumentation control [12,13]. In detail, the robotic surgical platform gives the operator the chance to reproduce human hand movements through instrument articulation and to see the operating field in 3 dimensions, allowing for a more precise surgery, especially in narrow spaces [32,33,34]. Since then, several studies compared robotic and laparoscopic approaches in different surgical gynecologic interventions, reporting conflicting findings [14,15,16,17,18].

A previous meta-analysis compared the outcomes of abdominal, laparoscopic, and robotic myomectomy; while a significant decrease was found in conversion rate, estimated blood loss (EBL), and postoperative bleeding in the robotic group, compared to the laparoscopic group, the difference in operating time was controversial [35]. In other studies, instead, longer operative times were reported for the robotic approach, due to the robot docking time [14,15,16,17,18,36].

Regarding hysterectomy for benign gynecological disease, similar post-operative outcomes were found in laparoscopic and robotic groups, except for a longer operative time in the robotic group [37,38]. Similar findings were reported by Sarlos et al. in a randomized controlled trial [39], while, in contrast, Lonnerfors et al. found a lower rate of post-operative complications and better short-term outcomes in the robotic group [40].

In a recent study, Kurt et al. compared the health-related quality of life (HRQoL) of women who underwent robotic surgery, laparoscopic surgery, and laparotomy for gynecological disease [41]. The authors found a higher HRQoL after robotic surgery, which might be due to an early ambulation, a faster recovery, and a decrease in hospital stay reported in the robotic group, compared with the laparotomy or laparoscopy groups [41].

Concerning EC, in a systematic review, Nevis et al. showed that EC women treated with robotic-assisted surgery had a reduction in EBL, compared to those treated with laparoscopic surgery, while no difference was found in the number of lymph nodes removed and complications rate [42]. In another study, no difference was found in oncological outcomes (i.e., overall and progression-free survival) among the surgical approaches [33].

Moreover, the outcomes of robotic and laparoscopic comparisons can also be affected by patients’ characteristics. For example, Seamon et al. found a significant 50% decreased risk of conversion in obese EC patients [26,43]. Regarding the detection rate, although obesity has been reported to decrease it because of the visceral adipose tissue that causes a more difficult visualization of lymphatic tissue [44], Bizzarri et al. found no difference in the bilateral detection rate during SLN mapping, regardless the higher prevalence of obese EC patients in the robotic group [25].

In this scenario, poor data are available in the literature about robotic–laparoscopic comparison in SLN biopsy during surgical treatment and staging of EC women. 2020 ESGO/ESTRO/ESP guidelines support SLN biopsy as an alternative to systematic lymphadenectomy to avoid its correlated morbidity [8,45]. In addition, a previous systematic review and meta-analysis demonstrated that SLN biopsy would be a valid alternative to systematic para-aortic and pelvic lymphadenectomy, not only in the surgical staging of low-risk groups of early-stage EC patients, but also in high-risk groups [9].

In the present study, we found no difference between laparoscopic and robotic approach for SLN biopsy in EC patients in each assessed outcome (i.e., SLN detection, intraoperative and postoperative complications, number of dissected SLN, and SLN identification and dissection). Such findings would indicate no superiority of an approach over the other one. Therefore, the choice among the two approaches for SLN biopsy could be based on the surgeon’ experience and skills.

However, despite the novelty of our study (i.e., it may be the first study about the topic in the literature), our findings might be affected by some limitations. Firstly, the low number of eligible studies (n = 2) did not allow us to calculate statistical heterogeneity, Secondly, the retrospective design of all included studies might limit the level of evidence of the findings. Thirdly, the two included studies did not assess all SLN biopsy outcomes; in particular, some outcomes were not assessed by any included study (e.g., estimated blood lost, length of hospital stay, and total intraoperative time), while other outcomes were assessed by only one of the two included studies (i.e., number of dissected SLN, intraoperative and postoperative complications, and conversion to laparotomy) [25]. Yet, comparison about some outcomes assessed only few cases. Fourthly, the outcomes considered in the included studies did not allow us to compare the safety of the two approaches for SLN biopsy, in terms of survival. Fifthly, we were unable to perform subgroup analyses based on the surgeon experience or patients’ characteristics that might affect SLN biopsy outcomes, such as body mass index, previous surgery, or International Federation of Gynecology and Obstetrics stage.

Based on these considerations, further studies with a prospective study design are encouraged to further investigate this field.

## 5. Conclusions

The laparoscopic and robotic approaches for SLN biopsy in EC patients do not appear to differ, in terms of SLN detection, intraoperative and postoperative complications, conversion to laparotomy, number of dissected SLN, and SLN identification and dissection time. Therefore, the choice between the two approaches could be based on the surgeon’s experience and skills. Further large and prospective studies are needed to further assess the topic.

## Figures and Tables

**Figure 1 jpm-13-00029-f001:**
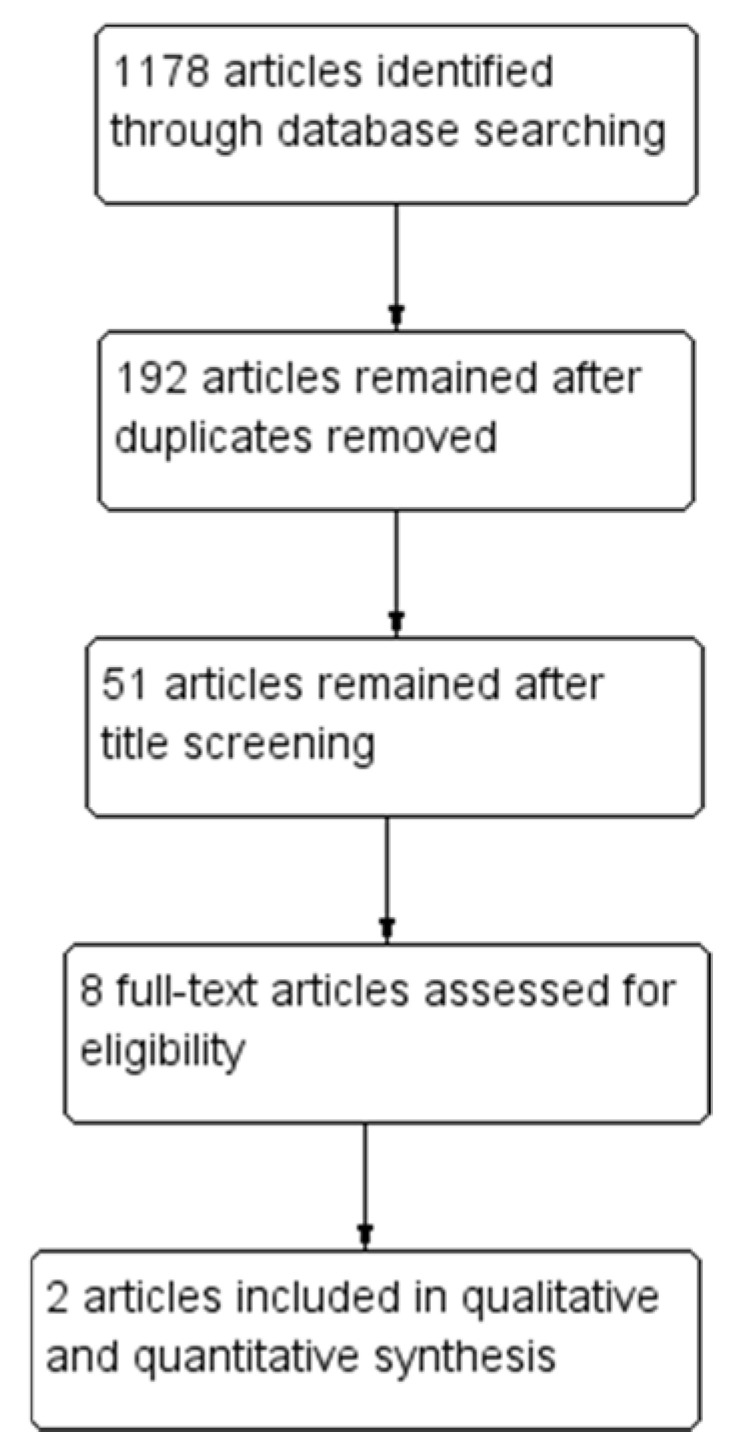
Flow diagram in the systematic review of the identified studies (PRISMA template (preferred reporting items for systematic reviews and meta-analyses)).

**Figure 2 jpm-13-00029-f002:**
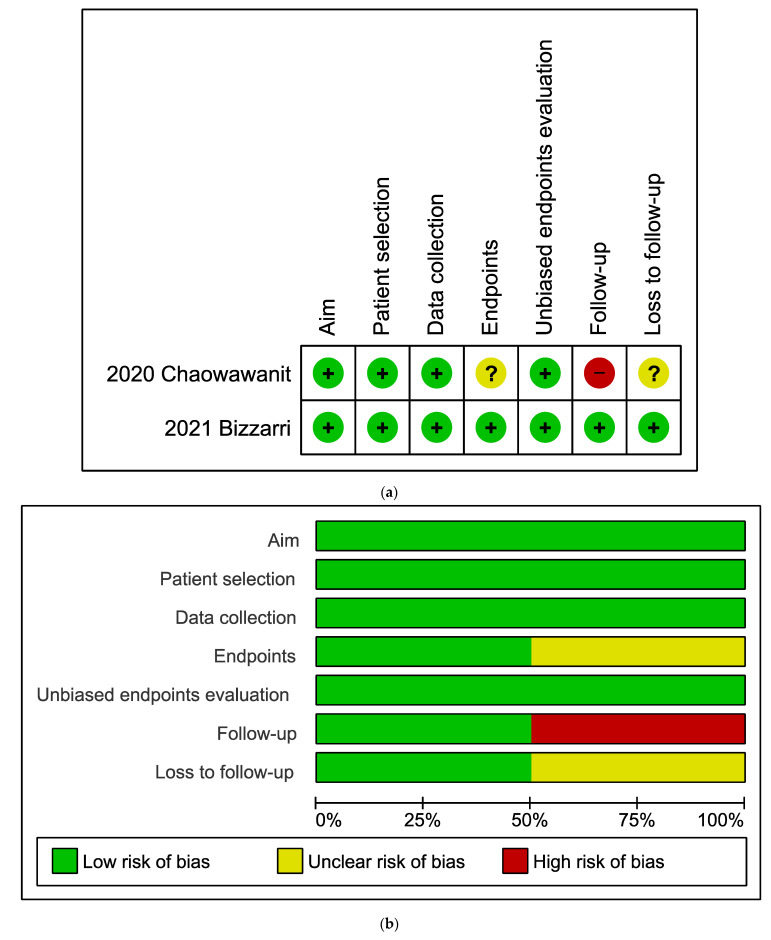
(**a**) Evaluation of the risk of bias within studies. Summary of the risk of bias in individual studies. Plus sign: low risk of bias; minus sign: high risk of bias; question mark: unclear risk of bias. (**b**) Risk of bias graph about risk of bias domains shown as percentages thorough all included studies.

**Figure 3 jpm-13-00029-f003:**
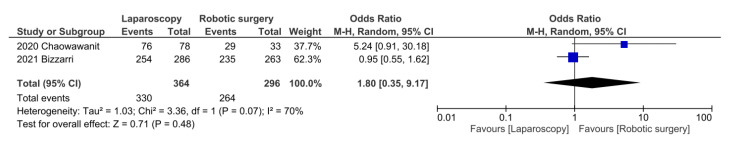
Forest plot of individual and pooled odds ratios, with 95% confidence interval (CI), for overall detection, comparing laparoscopy and robotic surgery for SLN biopsy in EC women.

**Figure 4 jpm-13-00029-f004:**
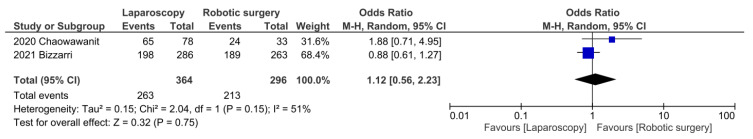
Forest plot of individual and pooled odds ratios, with 95% confidence interval (CI), for bilateral detection, comparing laparoscopy and robotic surgery for SLN biopsy in EC women.

**Figure 5 jpm-13-00029-f005:**
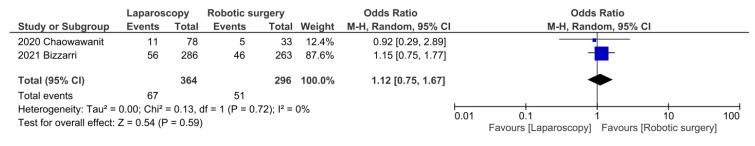
Forest plot of individual and pooled odds ratios, with 95% confidence interval (CI), for unilateral detection, comparing laparoscopy and robotic surgery for SLN biopsy in EC women.

**Figure 6 jpm-13-00029-f006:**
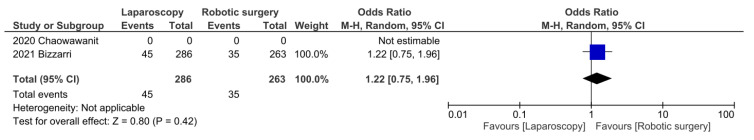
Forest plot of individual and pooled odds ratios, with 95% confidence interval (CI), for number of dissected SLN: 1, comparing laparoscopy and robotic surgery for SLN biopsy in EC women.

**Figure 7 jpm-13-00029-f007:**
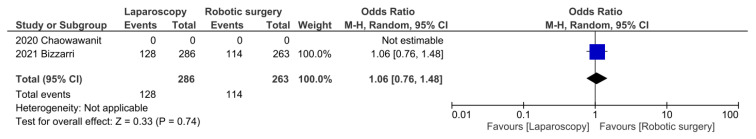
Forest plot of individual and pooled odds ratios, with 95% confidence interval (CI), for number of dissected SLN: 2, comparing laparoscopy and robotic surgery for SLN biopsy in EC women.

**Figure 8 jpm-13-00029-f008:**
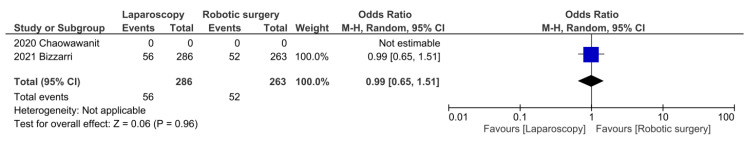
Forest plot of individual and pooled odds ratios, with 95% confidence interval (CI), for number of dissected SLN: 4, comparing laparoscopy and robotic surgery for SLN biopsy in EC women.

**Figure 9 jpm-13-00029-f009:**
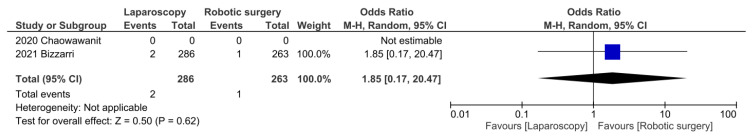
Forest plot of individual and pooled odds ratios, with 95% confidence interval (CI), for number of dissected SLN: 6, comparing laparoscopy and robotic surgery for SLN biopsy in EC women.

**Figure 10 jpm-13-00029-f010:**
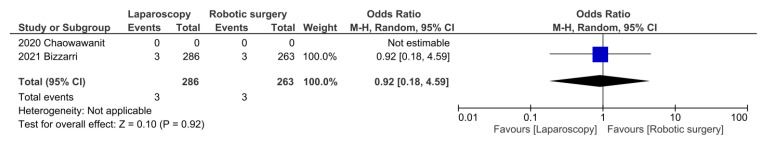
Forest plot of individual and pooled odds ratios, with 95% confidence interval (CI), for intraoperative complications, comparing laparoscopy and robotic surgery for SLN biopsy in EC women.

**Figure 11 jpm-13-00029-f011:**
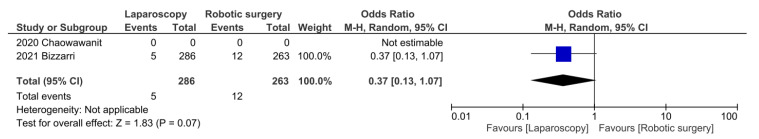
Forest plot of individual and pooled odds ratios, with 95% confidence interval (CI), for postoperative complications, comparing laparoscopy and robotic surgery for SLN biopsy in EC women.

**Figure 12 jpm-13-00029-f012:**
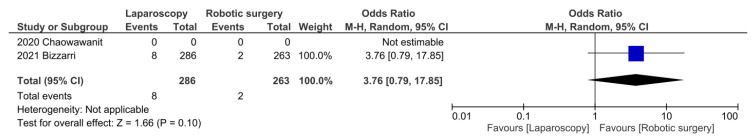
Forest plot of individual and pooled odds ratios, with 95% confidence interval (CI), for conversion to laparotomy, comparing laparoscopy and robotic surgery for SLN biopsy in EC women.

**Table 1 jpm-13-00029-t001:** Characteristics of included studies.

Study	Setting	Study Design	Study Period	Sample Size	Laparoscopic Surgery	Robotic Surgery
2020 Chaowawanit	Mater Hospital, Brisbane, Queensland, Australia	Retrospective, observational, cohort study	January 2017–May 2019	111	78	33
2021 Bizzarri	Fondazione Policlinico Agostino Gemelli IRCCS, Rome, Italy	Retrospective, observational, cohort study	January 2015–December 2019	549	286	263
Total				660	364	296

**Table 2 jpm-13-00029-t002:** Patients’ characteristics.

Study	Surgergical Approach	Age [Years] Mean ± SD (Range)	BMI [Kg/m^2^] Mean ± SD (Range)	Intraoperative Complications n (%)	Post Operative Complications n (%)	Prior Pelvic surgery n (%)	Deep Myometrial Invasion n (%)	FIGO Grade n (%)			FIGO Stage n (%)				Non-Endometrioid Histotype n (%)	LVSI n (%)	Identification Time [min] Mean (Range)	Dissection Time [min] Mean (Range)	Number of Lymph Nodes Median (Range)
								1	2	3	I	II	III	IV					
2020 Chaowawanit	Laparoscopic	62 ± 12	32.5 ± 7.1	nr	nr	39 (50)	19 (24.4)	51 (65.4)	11 (14.1)	7 (90)	68 (87.1)	1 (1.3)	8 (10.3)	1 (1.3)	9 (11.6)	16 (20.5)	36.4 (10–69)	18.5 (9–32)	1 (1–4)
	Robotic	63 ± 11	33.4 ± 7.5	nr	nr	18 (54.5)	7 (21.2)	22 (66.7)	5 (15.2)	3 (9.1)	31 (94.0)	2 (6.0)	0 (0)	0 (0)	3 (9.1)	7 (21.2)	40.9 (18–78)	15.9 (8–25)	1 (1–2)
2021 Bizzarri	Laparoscopic	61 (28–88)	26.0 (16.7–50)	3 (1.0)	5 (1.7)	nr	nr	38 (13.9)	162 (59.1)	74 (27.0)	227 (79.3)	19 (6.6)	37 (12.8)	3 (1.0)	57 (19.9)	77 (28)	nr	nr	2 (1–6)
	Robotic	64 (25–84)	34.8 (18.7–64.1)	3 (1.1)	12 (4.6)	nr	nr	21 (8.1)	190 (73.6)	47 (18.2)	204 (77.5)	20 (7.6)	37 (21.7)	2 (0.8)	35 (13.3)	76 (31.3)	nr	nr	2 (1–6)
Total	Laparoscopic	28–88	16.7–50	3 (1.0)	5 (1.7)	39 (50)	19 (24.4)	89 (25.9)	173 (50.5)	81 (23.6)	295 (81.0)	20 (5.5)	45 (12.4)	4 (1.1)	66 (18.1)	93 (25.5)	36.4 (10–69)	18.5 (9–32)	3 (1–6)
	Robotic	25–84	18.7–64.1	3 (1.1)	12 (4.6)	18 (54.5)	7 (21.2)	43 (14.9)	195 (67.7)	50 (17.4)	235 (79.4)	22 (7.4)	37 (12.5)	2 (0.7)	38 (12.8)	83 (28.0)	40.9 (18–78)	15.9 (8–25)	3 (1–6)

nr: not reported; BMI: body mass index; FIGO: International Federation of Gynecology and Obstetrics; LVSI: lymph vascular space invasion; SLN: sentinel lymph node.

**Table 3 jpm-13-00029-t003:** Details about indocyanine green injection and lymph node detection and mapping.

Study	Route of Surgery	ICG Injection				Identification Time [min] Mean (Range)	SLN Robotic Detection n (%)			Site of Mapping of Robotic First SLN n (%)						
		Concentration [mg/mL]	Dose [mL]	Site	Deepness [mm]		Unilateral	Bilateral	No Detection	External Iliac	Obturator	Internal Iliac	Common Iliac	Parametrial	Infra-mesenteric Para-Aortic	Presacral
2020 Chaowawanit	Laparoscopy	1.25	4	h 3 and h 9	10-4	36.1 (10–69)	11 (3.9)	65 (83.3)	2 (2.6)	81 (57.4)	31 (22.0)	9 (6.5)	16 (11.3)	1 (0.7)	2 (1.4)	1 (0.7)
	Robotic-assisted					40.9 (18–78)	5 (4.5)	24 (72.7)	4 (3.6)	37 (69.8)	7 (13.2)	1 (1.9)	8 (15.1)	0 (0)	0 (0)	0 (0)
2021 Bizzarri	Laparoscopy	1.25	1	h 3 and h 9	nr	10–15	56 (22.0)	198 (78.0)	32 (11.2)	297 (55.0)	181 (33.5)	34 (6.3)	18 (3.3)	nr	4 (0.7)	6 (1.1)
	Robotic-assisted						46 (19.6)	189 (80.4)	28 (10.6)	287 (59.9)	126 (26.3)	30 (6.3)	30 (6.3)	nr	2 (0.4)	4 (0.8)
Total	Laparoscopy	1.25	1-4	h 3 and h 9	10-4	nr	67 (18.4)	263 (72.3)	34 (9.3)	378 (55.5)	212 (31.2)	43 (6.3)	34 (5.0)	1 (0.1)	6 (0.9)	7 (1.0)

nr: not reported; ICG: indocyanine green; SLN: sentinel lymph node.

## Data Availability

Not applicable.

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
