# Peer review of "Comparison between Laparoscopic and Robotic Approach for Sentinel Lymph Node Biopsy in Endometrial Carcinoma Women"

_jpm, 2022, doi:10.3390/jpm13010029_

Round 1

Reviewer 1 Report

This is a well written systematic review and meta-analysis on a relevant topic: there is a robust search strategy and study selection, the study protocol is clearly documented and the result are clearly displayed. Unfortunately,  only 2 studies met the inclusion criteria: this highlights the efficient screening process from the Authors and leads to reflection about the overall quality of available study in literature on this topic. In would be very helpful if the Authors  could underline, in the Discussion paragraph, that the safety in oncological terms of  minimally invasive procedures must take into account not only the short-term outcomes but also the long-term outcomes such as disease free survival and overall survival. This may not be the specific focus of this meta-analysis, but in the post LACC trial era it is a necessary consideration

Author Response

REVIEWER #1

Comment #1

This is a well written systematic review and meta-analysis on a relevant topic: there is a robust search strategy and study selection, the study protocol is clearly documented and the result are clearly displayed.

Unfortunately, only 2 studies met the inclusion criteria: this highlights the efficient screening process from the Authors and leads to reflection about the overall quality of available study in literature on this topic.

  1. Authors response: We thank the Reviewer for the kind comments. We agree about the low number of included studies and reported it as a limitation in the “Discussion” section of our manuscript (lines 653-656 page 11). Unfortunately, they are the only available ones in the literature. We hope that our study might help in highlighting the need of further investigation in the field.
  2. Location: lines 293-296, page 11

Comment #2

In would be very helpful if the Authors could underline, in the Discussion paragraph, that the safety in oncological terms of minimally invasive procedures must take into account not only the short-term outcomes but also the long-term outcomes such as disease-free survival and overall survival. This may not be the specific focus of this meta-analysis, but in the post LACC trial era it is a necessary consideration

  1. Authors response: We thank the Reviewer for the kind remark. We agree about the importance of the long-term survival outcomes. Based on the outcomes considered in the included studies, we were unable to perform survival analyses and reported it as a limitation of our study.
  2. Location: lines 301-302, page 11

Reviewer 2 Report

The paper deals with an important medical problem, which is the introduction of new surgical technology in gynecologic oncology. Therefore, it deserves publication.
The disadvantage is that the meta-analysis is based on only two papers. It is a pity that at least the work of Nevis et al. was not included. As suggested by the authors, further research should be conducted in this area.
The statistical methods, for this stage, are very finicky.
Pg. 7 rows 185 - 191: Couldn't Figures 3, 4 and 5 as well as 6, 7, 8 and 9 be included collectively? This is because they are too fine-grained - e.g., additional Figure 9 is about comparing only two laparoscopic and one robotic cases. It is difficult to make comparisons on this basis.

Author Response

REVIEWER 2

Comment #1

The paper deals with an important medical problem, which is the introduction of new surgical technology in gynecologic oncology. Therefore, it deserves publication. The disadvantage is that the meta-analysis is based on only two papers. It is a pity that at least the work of Nevis et al. was not included. As suggested by the authors, further research should be conducted in this area.

  1. Author response: We thank the Reviewer for the kind comment. We agree about the low number of included studies and reported it as a limitation in the “Discussion” section of our manuscript (lines: 293-296, page 11). Unfortunately, they are the only available ones in the literature. In fact, based on our study selection criteria (i.e. inclusion criteria: peer-reviewed studies comparing laparoscopic and robotic approach in SLN biopsy in EC patients; exlusion criteria: studies in languages other than English video article, literature review, case reports), the study by Nevis et al. was not eligible for inclusion as it is a systematic review. However, we discussed findings from Nevis et al. study in the discussion section of our study (lines 267-271, page 11-12). We hope that our study might help in highlighting the need of further investigation in the field.
  2. Location: lines 293-296, page 11; lines 267-271, page 11-12

Comment #2

The statistical methods, for this stage, are very finicky.

  1. Author response: We thank the Reviewer for the kind words.
  2. Location: -

Comment #3

Pg. 7 rows 185 - 191: Couldn't Figures 3, 4 and 5 as well as 6, 7, 8 and 9 be included collectively? This is because they are too fine-grained - e.g., additional Figure 9 is about comparing only two laparoscopic and one robotic cases. It is difficult to make comparisons on this basis.

  1. Authors response: We thank the Reviewer for the suggestion. We tried to collectively include the Figures. Unfortunately, it reduced the readability; therefore, we left them as individual figures. Regarding the comparisons including a low number of cases, we agree that they can be difficult and we added it as an additional limitation of our study in the revised manuscript. On the other hand, we could not avoid to perform and graphically report them. Anyway, they were not statistically significant, further highlighting the need of additional investigation in the field.
  2. Location: lines 300-301, page 11

Reviewer 3 Report

General remarks:

A meta-analysis using only two publications limits the reliability of the findings.

The conclusions of the article add little to the conclusions of the two cited publications.

Since the issue of the journal is positioned as personalized medicine, it is necessary to provide personalized recommendations on the use of one or another method of taking a biopsy of the lymph nodes.

Author Response

REVIEWER 3

Comment #1

A meta-analysis using only two publications limits the reliability of the findings. The conclusions of the article add little to the conclusions of the two cited publications.

  1. Authors response: We thank the Reviewer for the kind comments. We agree about the low number of included studies and reported it as a limitation in the “Discussion” section of our manuscript (lines 293-296, page 11). Unfortunately, they are the only available ones in the literature. We hope that our study might help in highlighting the need of further investigation in the field.
  2. Location: lines 293-296, page 11

Comment #2

Since the issue of the journal is positioned as personalized medicine, it is necessary to provide personalized recommendations on the use of one or another method of taking a biopsy of the lymph nodes.

  1. Authors response: We thank the Reviewer for the comment. Since we found that laparoscopic and robotic approach for SLN biopsy in EC patients appear not differ in terms of SLN detection, intraoperative and postoperative complications, conversion to laparotomy, number of dissected SLN, and SLN identification and dissection time, we believe that the choice among the two approaches could be based on the surgeon’ experience and skills until new findings.
  2. Location: lines 310-312, page 11

Round 2

Reviewer 3 Report

I was satisfied with the response of the authors to my comments. The article will be useful for general oncologists who use fluorescent dyes to diagnose sentinel lymph nodes.